# N20D/N116E Combined Mutant Downward Shifted the pH Optimum of *Bacillus subtilis* NADH Oxidase

**DOI:** 10.3390/biology12040522

**Published:** 2023-03-30

**Authors:** Taowei Yang, Longze Pan, Wenhui Wu, Xuewei Pan, Meijuan Xu, Xian Zhang, Zhiming Rao

**Affiliations:** The Key Laboratory of Industrial Biotechnology, Ministry of Education, School of Biotechnology, Jiangnan University, Wuxi 214122, China

**Keywords:** water-forming NADH oxidase, pH shifting, NAD^+^ regeneration

## Abstract

**Simple Summary:**

Biocatalytic redox reactions perform a central role in producing chiral compounds, thus, have been receiving growing attention in the biochemistry area. The majority of redox reactions often require the nicotinamide cofactor as stoichiometric reductants. Various dehydrogenases have been employed for cofactor regeneration. However, the addition of extra substrates and the accumulation of corresponding byproducts demands extra costs for subsequent separation and purification. Water-forming NADH oxidase (Nox) has attracted substantive attention as it can oxidize NADH to NAD^+^ without concomitant accumulation of by-products. However, its applications have some limitations in some oxidation-reduction processes when its optimum pH is different from its coupled enzymes. Still, there are few reports regarding the shift of pH optima in Noxs. In this study, we modified the optimum pH of a Nox based on surface charge rational design. The Nox variations obtained in this work suggest promising properties for NAD^+^ regeneration.

**Abstract:**

Cofactor regeneration is indispensable to avoid the addition of large quantities of cofactor NADH or NAD^+^ in oxidation-reduction reactions. Water-forming NADH oxidase (Nox) has attracted substantive attention as it can oxidize cytosolic NADH to NAD^+^ without concomitant accumulation of by-products. However, its applications have some limitations in some oxidation-reduction processes when its optimum pH is different from its coupled enzymes. In this study, to modify the optimum pH of BsNox, fifteen relevant candidates of site-directed mutations were selected based on surface charge rational design. As predicted, the substitution of this asparagine residue with an aspartic acid residue (N22D) or with a glutamic acid residue (N116E) shifts its pH optimum from 9.0 to 7.0. Subsequently, N20D/N116E combined mutant could not only downshift the pH optimum of BsNox but also significantly increase its specific activity, which was about 2.9-fold at pH 7.0, 2.2-fold at pH 8.0 and 1.2-fold at pH 9.0 that of the wild-type. The double mutant N20D/N116E displays a higher activity within a wide range of pH from 6 to 9, which is wider than the wide type. The usability of the BsNox and its variations for NAD^+^ regeneration in a neutral environment was demonstrated by coupling with a glutamate dehydrogenase for α-ketoglutaric acid (α-KG) production from L-glutamic acid (L-Glu) at pH 7.0. Employing the variation N20D/N116E as an NAD+ regeneration coenzyme could shorten the process duration; 90% of L-Glu were transformed into α-KG within 40 min vs. 70 min with the wild-type BsNox for NAD^+^ regeneration. The results obtained in this work suggest the promising properties of the BsNox variation N20D/N116E are competent in NAD^+^ regeneration applications under a neutral environment.

## 1. Introduction

With the increasing attention paid to being greener and more sustainable, the application of biocatalysts has been growing in both research and industrial areas as their power is more widely appreciated [1,2,3]. Biocatalytic redox reactions perform a central role in producing chiral compounds with new functionalities and biofuels, thus, have been receiving growing attention in the biochemistry area [4,5,6]. Nicotinamide adenine dinucleotide (NAD^+^) is an essential molecule involved in a vast range of biological processes [7]. It was discovered as the first “coenzyme” over 100 years ago. More than 1500 enzymatic reactions in microorganisms require NAD(H) and NADP(H) as cofactors, which critically influence redox balance and cellular metabolism [8]. Therefore, cofactor regeneration and carbon metabolism must be coupled to make feedstock produce high-value-added chemicals more efficiently [8]. However, the high price and rather poor stability of pyridine nucleotide cofactors do not conform to the economic regulations of industrial-scale processes. Thus, cofactor regeneration is indispensable to avoid the addition of large quantities of NADH or NAD^+^ in enzyme-catalyzed reactions [4,8]. Various dehydrogenases, such as alcohol dehydrogenase [9], lactate dehydrogenases [10], glucose dehydrogenase [11] and formate dehydrogenase [12], are widely used in cofactors regeneration. However, the addition of extra substrates for cofactors regeneration and the accumulation of corresponding byproducts demands extra costs for subsequent separation and purification [4,13].

NADH oxidases (Noxs), which catalyze the oxidation of NADH by reducing molecular O_2_ to either H_2_O_2_ or H_2_O, show a promising candidate for cofactor NAD^+^ regeneration [13,14]. Compared to H_2_O_2_-forming Noxs, which can cause enzyme deactivation, water-forming Noxs have attracted substantive attention in a wide range of industrial enzymatic reactions as they can oxidize cytosolic NADH to NAD^+^ without by-products produced [15]. A number of H_2_O-forming Noxs have been identified in lactic acid bacteria [15,16,17,18] and *Streptococcus* sp. [19,20]. Among these Noxs, most have been shown as promising candidates for cofactor NAD^+^ regeneration. Shi et al. [21] found that overexpression of a water-forming NADH oxidase could enhance the ethanol production in *Saccharomyces cerevisiae* BY4741 and the butyric acid production in *Clostridium acetobutylicum* 428 (Cac-428) under anaerobic conditions. It was found that combined glycerol dehydrogenase and NADH oxidase for in situ NAD^+^ regeneration increases the conversion of glycerol to 1,3-dihydroxyacetone [22,23]. Zhang et al. [24] introduced an NADH oxidase into *E. coli* cells to recycle NAD^+,^ which improved L-gulose production. Hong et al. [25] observed that the introduction of the GabTD/Nox system remarkably diminished the demand for NAD^+^ and increased glutaric acid production from lysine. Su et al. [26] used galactitol dehydrogenase coupled with water-forming NADH oxidase achieving an efficient enzymatic synthesis of L-tagatose. Cui et al. [27] constructed a two-enzyme system containing 2,3-butanediol dehydrogenase and H_2_O-forming NADH oxidase which resulted in optically pure acetoin formation without introducing any by-product.

In a previous study, we characterized an H_2_O-forming Nox (BsNox) from *Bacillus subtilis*, and the BsNox was introduced to redistribute the carbon flux to acetoin by manipulating NAD^+^ levels [28]. We also noticed that the maximum activity of BsNox was observed at pH 9.0 [28]. Nox is known to play a key role in cofactor regeneration for NAD^+^-dependent enzyme reactions; however, its industrial applications have some limitations in specific cases when its optimum pH is different from its coupled enzymes [29]. Nonetheless, as far as we know, there is little information about modifying the optimum pH of Nox. Surface engineering of proteins has been demonstrated to be a valuable tool to yield more stable, active, and soluble proteins [30]. The Rosetta approach varies net charge by adjusting reference energies of the positive or negatively charged residue types when scoring protein sequences and conformations [31]. Kim et al. [32] achieved to shift of the pH optima of *Aspergillus niger* PhyA Phytase to match the stomach condition by substituting amino acids in the substrate-binding site with different charges and polarities. Pokhrel et al. [33] successfully shifted the pH optimum of a *Bacillus circulans* xylanase by introducing charged residue. Ma et al. [34] constructed an elaborate GH11 xylanase database with pH annotation, and developed a data driven protein engineering strategy to redesign the pH adaptation of xylanase. In a previous study, we also successfully modified the optimal pH of an aspartase from *Bacillus* sp. YM55-1 based on surface charge rational design, and improved its efficiency for β-aminobutyric acid production [35].

In this study, to make the properties of the BsNox suitable for its applications in a neutral environment, protein engineering such as site-directed mutagenesis is carried out based on surface charge rational design. Rosetta software was used to calculate and design the surface charge of the BsNox. The usability of the BsNox and its variations for NAD^+^ regeneration in a neutral environment were observed by coupling a glutamate dehydrogenase with BsNox for α-ketoglutaric acid (α-KG) production at pH 7.0.

## 2. Materials and Methods

### 2.1. Materials

The bacterial strains, plasmids, and primers used in this study are listed in Appendix A. *E. coli* was cultured in Luria-Bertani (LB) medium at 37 °C on a rotary shaker at 180 rpm, and 50 μg/mL of kanamycin was supplemented if necessary.

### 2.2. Site-Directed Mutagenesis

Site-directed mutagenesis of the *BsNox* gene was carried out using whole-plasmid two-step PCR with the primers (Appendix A) containing required codons for mutations (sense strand) and the antisense strand based on the expression vector pETDuet-1 [36]. DNA sequencing for the mutated *BsNox* genes was conducted by Shanghai Sangon Biotech Co. Ltd. (Shanghai, China).

### 2.3. Enzyme Preparation, Purification, and Identification

The cell pellets were suspended and washed with 0.1 M potassium phosphate buffer (pH 7.0) 3 times. For the preparation of crude Nox, cells were resuspended in a washing buffer containing 0.15 mM FAD and 62.5 mM L-cysteine-HCl. DNase (2000 units) and phenylmethyl sulfonyl fluoride (1 mM) were then added to the suspension. Enzyme purification was performed as described by Zhang et al. [28]. The recombinant Nox was expressed as a His6-tagged protein in *E. coli* and purified first by affinity chromatography on a Ni-NTA Sepharose prepacked column HisTrap HP (GE Healthcare, Uppsala, Sweden). The pooled fractions were then loaded on a Superdex TM 200 (10/300GL) equilibrated with the buffer (20 mM Tris–HCl (pH 8.0) and 150 mM NaCl) using an ÄKTA Protein Purifier system (Pharmacia, Uppsala, Sweden) [37]. The purified protein samples were analyzed with SDS-PAGE using 12% polyacrylamide gels and visualized with 0.25% Coomassie Brilliant Blue G-250.

### 2.4. Activity and Optimum pH Determination

Nox activity was determined by a photometer assay at 340 nm, as described by Gao et al. [15]. Each reaction mixture contained 0.3 mM EDTA, 0.05 mM FAD, 0.3 mM β-NADH, a certain amount (~3 μg/mL) of enzyme, and 50 mM phosphate buffer (pH 7.0) in a total volume of 1.5 mL. The assay was carried out at 37 °C, with 1 U of activity corresponding to the oxidation of 1 μmol of NADH per minute.

The optimal pH for enzymes was determined by measuring the enzyme activity at pH 7.0–9.0, and the residual enzyme activity was measured under standard assay conditions.

### 2.5. Catalytic Performance of Enzymes Mixture

Enzymatic synthesis of α-ketoglutaric acid (α-KG) was performed in a 100 mL catalytic system containing 100 mM Tris-HCl (pH 7.0), 10 mM L-glutamate, 1 mM NAD^+^, 2 U/mL glutamate dehydrogenase and 2 U/mL Nox at 37 °C, and 200 rpm under a batch system in flasks.

The concentration of α-KG was measured via high-performance liquid chromatography (HPLC) equipped with an HPX-87H column (BioRad) at 40 °C with a flow rate of 0.5 mL/min. The mobile phase gradient was 0.5 mM H_2_SO_4_. A UV detector (model LC-9A; Shimadzu, Kyoto, Japan) was used for detection at 338 nm. L-glutamate titer was determined by a biological sensing analyzer (SBA, Shandong, China). Triplicate measurements were made for each sample, and the corresponding standard errors were calculated.

### 2.6. ROSETTA Supercharge, Modeling and Molecular Dynamics Simulation

ROSETTA is a unified software package for protein structure prediction and functional design based on full computer support [38]. Rosetta supercharge considers the interactions between the amino acids, including electrostatic interactions, hydrogen bonding, and hydrophobic interactions, which is flexible enough to allow the hydrophobic amino acids to be mutated to adjust the protein surface potential distribution. Thus, the Rosetta approach can preserve and potentially add stabilizing interactions on the surface while increasing net charge. The Rosetta supercharging protocol uses the score 12 full-atom energy function. The major terms of the full-atom energy function are Lennard-Jones attraction, Lennard-Jones repulsion, an implicit solvation model disfavoring burial of polar groups, hydrogen bonding, a statistical residue pair term for electrostatics, side-chain rotamer probability, and a reference energy used to favor a native-like abundance of each amino acid type. The Rosetta approach varies net charge by adjusting reference energies of the positive or negatively charged residue types. Users can specify a net target charge, and the protocol will iteratively increment the charged-residue reference energies until the desired net charge is achieved [31]. The three-dimensional structures of the final mutants were modeled using the BsNOX structure with PDB Id 3ge6 as the template. The models were created using the SWISS-MODEL Workspace. The potential changes in enzyme surface were analyzed by PYMOL. The molecular dynamics (MD) simulation using the MD simulation package GROMACS v. 2018.4.

## 3. Results

### 3.1. Screening of Residues on the Surface of BsNox for Site-Directed Mutagenesis

The initial sequence of gene *yodC* (GeneID:939506) encoding NADH oxidase (BsNox) form *B. subtilis* 168 was submitted to SWISS-MODEL Workplace for modeling, and the BsNox structure was obtained. In this work, 3ge6 with a sequence identity of 43.5% was used as the template, which is at 1.85 Å resolution. According to the current definition, if the sequence homology of the protein to be modeled is more than 40% after comparing it with the template sequence, the three-dimensional structure of the protein can be predicted by the method of homologous protein modeling. In this result, the value of GMQE is 0.80 (The confidence range of GMQE is 0–1, with a higher value indicating better quality) [39]. The value of QMEAN is 0.76 ± 0.05 (the range of QMEAN is −4–0; the closer to 0, the better the matching degree between the tested protein and the template protein will be). So we think that the structure’s quality is sufficient for our further work. Then PYMOL was used to generate the potential surface map of the protein, and the total surface charge of the BsNox was determined to be −10. For rational quantification, we specified a net target charge as −50 (the total surface charge was set from −10 to −50) and then iterated the protocol. According to the output results, as shown in Figure 1, fifteen sites were selected, which contain nonpolar residues (A118D, I168D), uncharged polar residues (T2D, N20D, Y66D, Q68E, Q114E, N116E, Y119E, Q189D), and polar charged residues (K25D, K85D, R171E, H188E, K197D). These mutants were expected to have lower pH optimum than the wild type. As shown in Figure 1, the predicted mutation sites are located on the surface of the BsNOX structure. Taking the double mutant N20D/N116E as an example, after mutating into acidic amino acids, the initial potential changes of both residues 20 and 116 are lower than that of the WT (Figure 2).

### 3.2. Expression of BsNox and Its Mutants

The *BsNox* gene and its mutagenic genes were separately cloned to the plasmid pETDuet-1 and expressed in *E. coli* BL21. In order to confirm the successful expression of BsNox and its variants, SDS-gel analysis was used with relevant cell lysates. After 12 h of induction with IPTG, as shown in Figure 1, thirteen mutants were all successfully over-expressed in *E. coli* BL21. The bands of proteins are clear, and the molecular weight of BsNox is about 22.3 kDa (Figure 3). Proteins were purified using Ni-NTA resins as described in our previous study [37] and isolated at high purity (Appendix A).

### 3.3. pH Profile and Stability

The pH-activity profiles of the BsNOX mutants were determined and compared with that of the wild-type. As shown in Figure 4, among the twelve mutants, T2D, N20D, K25D, Y66D, N116E, A118D and I168D showed a higher activity at pH 7.0, and T2D, N20D and N116E also had a higher activity at pH 8.0. Significantly, the substitution of the asparagine residue (N116) with a glutamic acid residue (N116E) showed the best effect among the single-site mutants, and its optimum pH shifts from 9.0 to 7.0, with an increase in activity by 61.2% at pH 7.0 and 47.6% at pH 8.0. However, compared to the wild-type, the N116E mutant showed a decrease in activity by about 15% at pH 9.0, while the N20D mutant had a similar activity at pH 9.0. Thus, N20D/N116E combined mutant was constructed, and it showed a pronounced effect on the pH-activity profile ranging from pH 7.0 to pH 9.0. The pH optimum of the N20D/N116E mutant moved to neutral (pH 7.0) (Figure 4). The double mutant not only down-shifted the pH optimum of BsNox but also significantly increased its specific activity, which was about 2.9-fold at 7.0, 2.2-fold at 8.0 and 1.2-fold at 9.0 that of the wild-type.

The activities of the double mutant N20D/N116E under a wide range of pH from 5 to 10 were furtherly investigated to test its applicability in some specific processes. The results are summarized in Figure 5, which clearly shows that N20D/N116E displayed a higher activity within a wide range of pH from 6 to 9, which was wider than the wide type; it decreased significantly only when pH was outside this range, especially under acid conditions (pH < 6).

Furthermore, the pH stability of the BsNox and its variants was observed. All purified enzymes were stored at 4 °C for 24 h, pH range from 7 to 9, and tests were carried out at 37 °C in 50 mM potassium phosphate buffer. The results in Figure 6 show that the enzyme characteristics of the wild-type and its mutants were apparently similar from pH 7.0 to 9.0; the activity for all recombinants is more stable, especially the variant N116E which showed more stability when heat treatment was applied to the enzyme for 60 h.

### 3.4. Molecular Dynamics Simulation

GROMACS is a specific software for molecular dynamics simulations. The molecular dynamics simulations show that the RMSD value of variant N20D/N116E was similar to the wild-type BsNox (Figure 7a). The Root Mean Square Fluctuation (RMSF) values of mutants were calculated to assess structural modifications to the wild-type. Figure 7b shows that double mutant (N20D/N116E) increased the RMSF value between 12 and 35 sites, and between 90 and 115 sites, the highest RMSF value reached 0.71 Å.

Figure 8a shows that both mutant residues N20D and N116E are far away from the catalytic residue Cys152 located in the deepest part of the FAD-binding channel. Most water-forming NADH oxidases have typical Rossman folds, and each Rossman folding domain has a binding region of substrate NADH or cofactor FAD [40]. Figure 8b shows that the N20D residue is located on the loop of the outer side of Rossman’s folded domain. The substitution of the asparagine residue with a glutamic acid residue in the 116 residue widened the gate of the FAD-binding channel (Figure 8c,d).

### 3.5. Using Mutant N20D/N116E for NAD^+^ Regeneration Enhanced α-Ketoglutarate Production

Glutamate dehydrogenase catalyzes the conversion of L-Glu to α-KG with concomitant reduction of NAD^+^ to NADH. For enzymatic bio-oxidation of L-glutamic acid (L-Glu) to α-ketoglutaric acid (α-KG), the continuous consumption of cofactor NAD^+^ strikingly increases the cost of the process. To avoid the addition of large quantities of cofactor NAD^+^ in this process and observe the usability of the BsNox and its variants for NAD^+^ regeneration, in this work, an efficient and environmentally friendly enzymatic system containing a glutamate dehydrogenase mutant K218D (GDH) obtained previously [41] for L-Glu oxidation and BsNox for NAD^+^ regeneration was designed for α-KG biosynthesis under neutral environment. As shown in Figure 9, the reaction mixture with BsNox and its variant N20D/N116E could convert L-Glu into α-KG fast, while the control (without BsNox addition) had negligible efficiency. The variant N20D/N116E increased NAD^+^ regeneration ability, thus improving the consumption rate of L-Glu by 42.8% at 45 min, compared with the wild-type BsNox. The highest conversion rate was recorded with variant N20D/N116E as the NAD^+^ regeneration coenzyme; 90% of L-Glu was consumed within 40 min with ~9 mM α-KG produced. At the same time, we noted that it would take a much longer time (70 min) to consume 90% of L-Glu with wild-type Nox for NAD^+^ regeneration. Employing the variation N20D/N116E as an NAD+ regeneration coenzyme could shorten the process duration.

## 4. Discussion

Water-forming NADH oxidase (Nox) has attracted substantive attention in a wide range of industrial enzymatic oxidation-reduction reactions as they can oxidize NADH to NAD^+^ only with H_2_O produced [15]. Nox has been widely employed to couple with certain dehydrogenases, which successfully enhance many important chemicals’ biosynthesis. Glutamate dehydrogenase (Gdh) catalyzes the conversion of L-Glu to α-KG with concomitant reduction of NAD^+^ to NADH. Thus, it is an efficient and environmentally friendly system by coupling a Gdh for L-Glu oxidation and a Nox for NAD^+^ regeneration for the enzymatic synthesis of α-KG. Because it can avoid the addition of large quantities of cofactor NAD^+^ without by-products produced. In our previous study, we characterized a glutamate dehydrogenase (BsGdh) from *B. subtilis* 168, and it was found that it has relatively high activity under neutral pH for conversion of L-Glu to α-KG but very little activity at pH 9.0 [41]. Previously, we characterized an H_2_O-forming Nox (BsNox) from *Bacillus subtilis* [28]. However, we also noticed that the maximum activity of BsNox was observed at pH 9.0, and its specific activity was depressed when pH was under neutral conditions [28]. Therefore, for efficient biosynthesis of α-KG through the promising system, it is of considerable importance and mandatory to modify the BsNox pH tendency through rational design and site-directed mutation. Nonetheless, as far as we know, there are few reports regarding the shift of pH optima in Noxs.

Studies have shown that engineering the surface charge of enzymes can modify their characteristics, such as optimal pH, and enzyme catalysis [30]. Previously, we successfully modified the optimal pH of BsGdh based on surface charge rational design, and the optimal pH of K218D was reduced from 8.0 to 7.0, with an increase of activity by 5.4-folds [41]. In this study, it is important to note that substitution of this uncharged residue (Asn) with polar negatively charged residues (Asp/Glu) could shift pH optima from 9.0 to 7.0 in Noxs. The molecular dynamics simulations show that the RMSD value has little difference between variant N20D/N116E and the wild-type BsNox (Figure 7a), indicating that variant N20D/N116E had little effect on the stability of the protein structure. Most water froming NADH oxidases have typical Rossman fold, and each Rossman folding domain has a binding region of substrate NADH or cofactor FAD [40]. In this BsNox structure, N20D residue is located on the loop of the outer side of Rossman’s folded domain. The Root Mean Square Fluctuation (RMSF) values of mutant N20D increased (Figure 7b), indicating that the flexibility of the residue increase probably contributed to the increase of the mutant activity. The cofactor FAD plays a significant role in Nox oxidative activity [42]. Furthermore, the substitution of the asparagine residue with a glutamic acid residue in the 116 residue led to the weakening of the hydrophobic interaction between residues near the gate of the FAD-binding channel and resulted in the widening of the channel gate, making it easier to contact each other among FAD, substrate and active site; thus increased the mutants specific activity. The engineering of enzyme charges to shift the pH-activity profile by modulating the interaction with the active site residues has been studied by some researchers. Kim et al. [32] achieved a shift in the pH optima of *Aspergillus niger* PhyA Phytase by substituting amino acids in the substrate-binding site with different charges and polarities. Pokhrel et al. [33] successfully shifted the pH optimum of a *B. circulans* xylanase by introducing charged residue. In a previous study, we also successfully modified the optimal pH of an aspartase from *Bacillus* sp. YM55-1 based on surface charge rational design [35]. The shift in pH optimum was probably due to charge repulsion, as explained by Pokhrel et al. [33], the charge repulsion may be direct, between the Asp/Glu inserted and the catalytic sites, or it may be indirect.

## 5. Conclusions

Surface charge-based rational design of a *B. subtilis* NADH oxidase by introducing negatively charged residues could downshift its pH optima from 9.0 to 7.0. Combing N20D/N116E mutant is more feasible to downshift the BsNox optimum pH from an alkaline condition to a neutral one. Its usability for NAD^+^ regeneration was demonstrated by coupling N20D/N116E variant with a glutamate dehydrogenase with a higher α-KG productivity from L-Glu at pH 7.0. Employing the variation N20D/N116E as an NAD^+^ regeneration coenzyme could shorten the process duration. The results obtained in this work suggest that the promising properties of the BsNox variation N20D/N116E are competent in NAD^+^ regeneration applications under a neutral environment.

## Figures and Tables

**Figure 1 biology-12-00522-f001:**
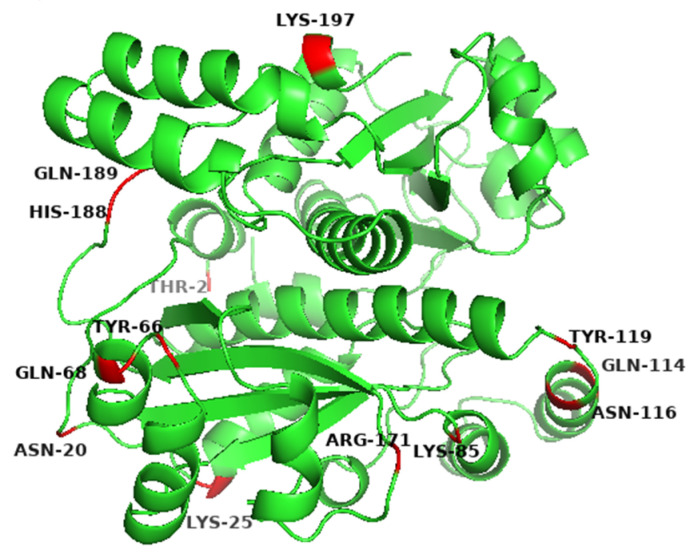
Predicted mutation sites in the wide-type BsNOX structure.

**Figure 2 biology-12-00522-f002:**
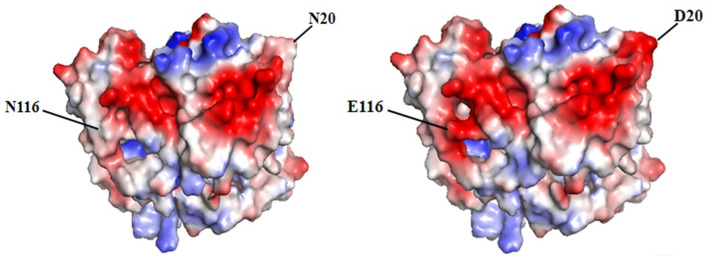
Potential changes on the enzyme surface. Acidic amino acids are shown in red, and alkaline amino acids are shown in blue. Left: the wild-type BsNOX. Right: the N20D/N116E variant.

**Figure 3 biology-12-00522-f003:**
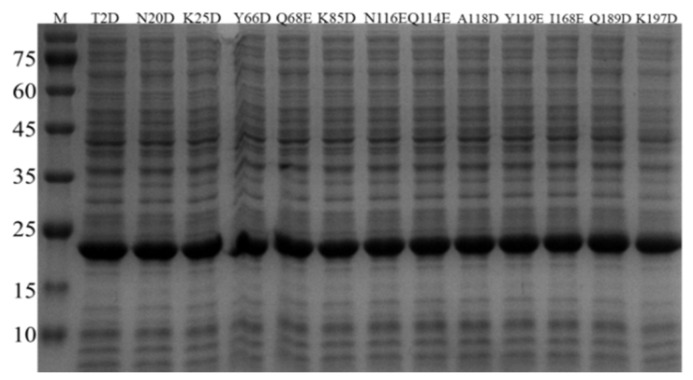
Protein expression analysis of variants on SDS-PAGE. The cell lysate was examined by SDS-PAGE under denaturing conditions, using the prestained marker (Takara, Japan) as the reference.

**Figure 4 biology-12-00522-f004:**
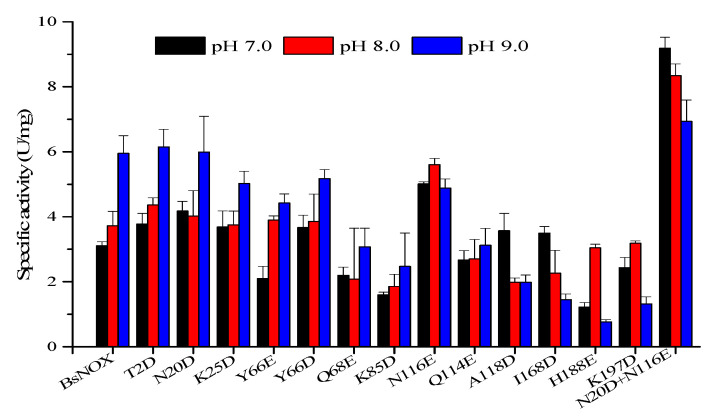
Effects of pH on the specific activity of BsNox and its variants.

**Figure 5 biology-12-00522-f005:**
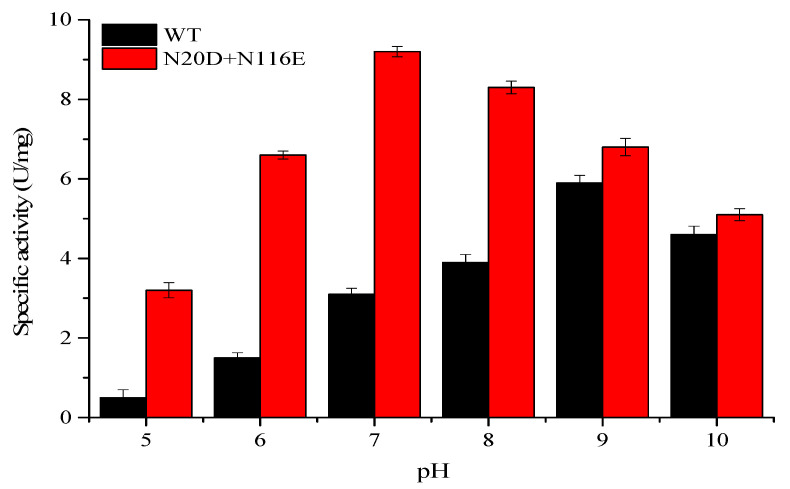
Effects of pH on the activities of WT and double mutant N20D/N116E.

**Figure 6 biology-12-00522-f006:**
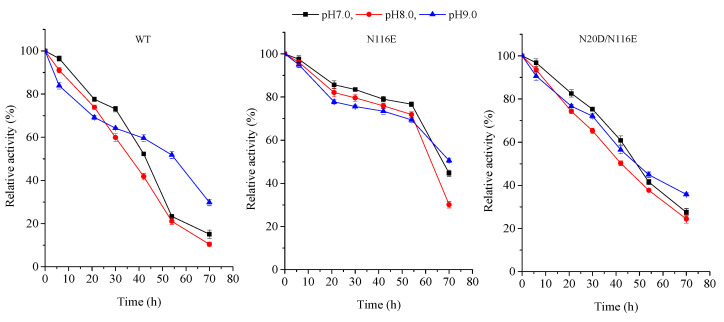
pH stability of BsNOX-WT and its mutants.

**Figure 7 biology-12-00522-f007:**
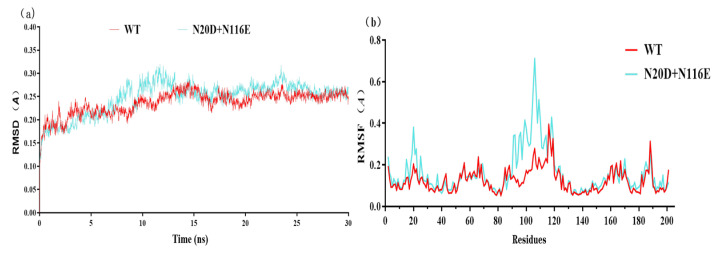
Molecular dynamics simulations of wild-type BsNox and its variant N20D/N116E. (**a**) The RMSD values of WT and its variant N20D/N116E. (**b**) The RMSF values of WT and its variant N20D/N116E.

**Figure 8 biology-12-00522-f008:**
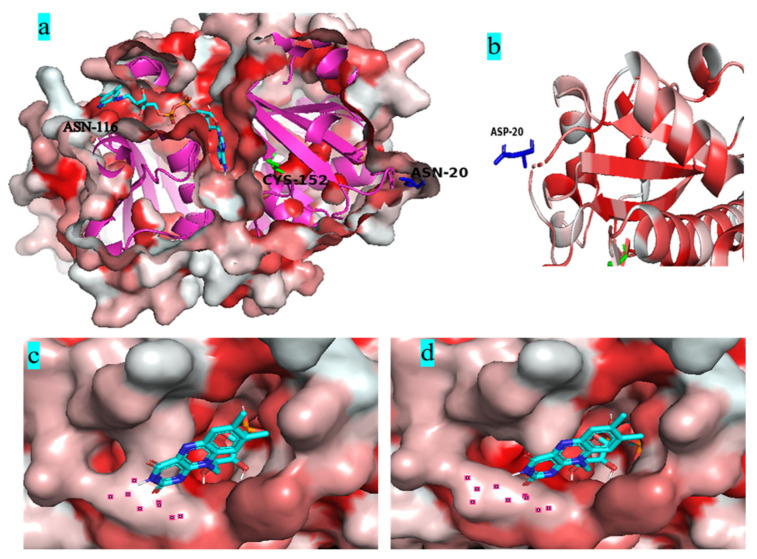
Molecular dynamics simulations of Nox and its variants. (**a**): the generation of the Nox structure, showing the cross-section of the FAD-binding channel and the position of active residue Cys152, the selected residues N20 and N116; (**b**): The position of residues 20 in NOX; (**c**), The surface region of WT closed to the channel of FAD; (**d**), The surface region of N20DN116E closed to the channel of FAD; In figure (**c**,**d**), the red regions represent hydrophobic residues and the white are hydrophilic residues. The darker the color is, the stronger the hydrophobicity is.

**Figure 9 biology-12-00522-f009:**
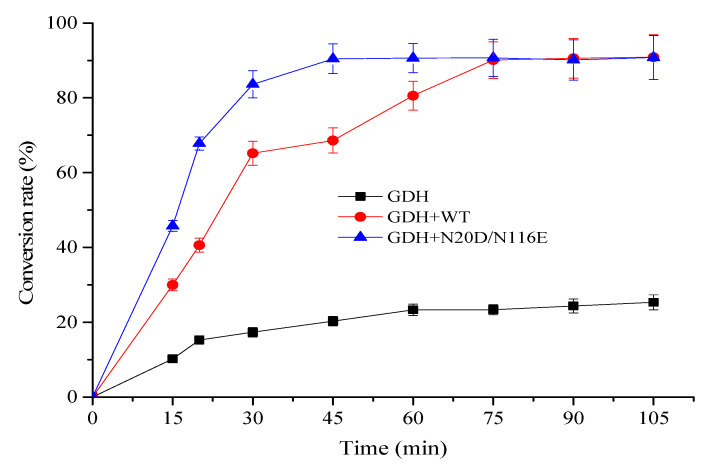
Time profile of conversion of L-Glu into α-KG by coupling GDH with NOXs.

## Data Availability

The data that support the findings of this study are available from the corresponding author upon reasonable request. All datasets generated for this study are included in the article/Appendix A.

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
