# Peer review of "N20D/N116E Combined Mutant Downward Shifted the pH Optimum of Bacillus subtilis NADH Oxidase"

_biology, 2023, doi:10.3390/biology12040522_

Round 1

Reviewer 1 Report

The  authors describe the general method they employed to determine which point mutations to make at which positions, but I believe more detail is needed here. Why are these certain mutations recommended at these positions? More detail from how Rosetta supercharge works is needed.  Without this information, it would appear that Rosetta is a black box.

Line 51: grammatical error, change “have been employing” to “employ”

Line 267, 268: There is no continuity from the previous sentence to this sentence.

The paper can be accepted after minor revision in English and providing additional detail on the experimental design and rationale for the targets selection.

Author Response

The  authors describe the general method they employed to determine which point mutations to make at which positions, but I believe more detail is needed here. Why are these certain mutations recommended at these positions? More detail from how Rosetta supercharge works is needed.  Without this information, it would appear that Rosetta is a black box.

Response: More detail from how Rosetta supercharge works has been listed in section 2.6. Please see line 151-166 in the revised manuscript. For rational quantification we specify a target net charge as -50 (the total surface charge was set from -10 to -50) and then iterate the protocol. According to the output results, fifteen sites were selected. Thanks for your good suggestion.

Line 51: grammatical error, change “have been employing” to “employ”

Response: Done and thanks.

Line 267, 268: There is no continuity from the previous sentence to this sentence.

Response: We have revised those sentences, and thanks for your valuable suggestion.

The paper can be accepted after minor revision in English and providing additional detail on the experimental design and rationale for the targets selection.

Response: Additional detail on the experimental design and rationale for the targets selection has been listed in section 2.6. We also have tried our best to polish the English to make it more readable. Thanks for your valuable suggestion.

Those criticisms are all valuable and very helpful for revising and improving our paper. We appreciate very much for your efforts on reviewing this manuscript.

Reviewer 2 Report

The manuscript describes the generation of a double mutant version of the water-forming NADH oxidase BsNox. A total of 15 different versions of the enzyme were generated and their activity compared at pH 7.0, 8.0 and 9.0. The two most promising variants were combined in a double mutant that showed higher activity and stability at pH 7.0, thus increasing the industrial usability of the oxidase. Its usability was demonstrated using the generation of α-ketoglutarate with the glutamate dehydrogenase as example.

The manuscript would benefit from thorough English editing. Some words are used in the wrong context and some sentences incomplete and/or hard to understand.

To start with, all figure legends, including the one of the supplementary figure, are poor. They are far too short and do not give the reader the information necessary to understand and interpret the figure. For example, it remains completely unclear what exactly is depicted in the supplementary figure 1. Are these the charges of the wildtype enzyme? Or of one of the variants? And which of the three subfigures of figure 3 depicts which enzyme variant? What is GDHK218D in figure 5? And why is that never mentioned in the text?

The material and method part is missing all information on the bioinformatics that are the basis for the choice of the variants. Please include these. In there, also include which enzyme/protein was used as a model for the generation of your BsNox structure. Give a short overview on why you determined the structure’s quality to be sufficient for your further work. I would also strongly suggest to mark the amino acids chosen for mutation in the figure with the structure to give the readers an idea of where these can be found in the protein: active site or surface? This should also be a basis for a part of your discussion where you should elucidate why certain modifications show certain effects – something that is now missing completely in the manuscript.

Please add the gel containing the purified enzyme variants in your supplement to give the reader the possibility of noting their purity.

Not necessarily needed, but it might be interesting to test even lower pH values for the activity of your double mutant as we see a clear increase in activity from pH 9 to 7 in figure 2.

Ll. 203 to 205: In my opinion, the conclusion that the highest RMSF value indicates the increased flexibility of the residue and thus explains the increase in activity is too superficial. Here, as mentioned above in the second paragraph, the you should refer to the position of the relevant amino acids in the protein and also compare where the two top mutants are found. The highest RMSF value, after all, coincides with neither of them. All of this, of course, belongs in the discussion part of the article.

The same goes for the conclusion based on the change in pI, which is depicted in ll. 211-212. As the pI of about 5.0 is far away from the pH values tested, this seems like a bit of a stretch and the hypothesis needs to be explained in more detail.

Figure 5: A comparison to the conversion rates at the former optimal pH 9.0 is missing.

The discussion is lackluster. The first paragraph is more fitting as an introduction as no new information is referenced and should indeed be integrated there. Large parts of the second paragraph have also been mentioned earlier in the document (ll. 258-267). Please discuss your results instead of simply repeating information, including the position of the amino acids that were mutated in your study and why these could have the influences determined as mentioned above. Right now, the second part of the discussion reads more like the conclusion, which is obvious as both parts are relatively identical.

Last but not least, the authors keep mentioning the need for a Nox with a neutral pH optimum, claiming its usability for ‘some specific industrial processes’. I would strongly suggest the mention of at least one specific example for which this change is mandatory.

Some minor points:

-        L. 104: The medium is called Luria-Bertani.

-        Please make sure to use a space between numbers and units everywhere.

-        L. 115: What does PPFC stand for?

-        The purification of the enzymes in 2.3 is described very superficially. This would be sufficient if a reference that includes all details needed for reproduction was added, but as no reference is given, please include enough details to ensure the reproducibility.

-        L. 164: What is protein glue?

-        L. 174: The increase in activity at pH 8.0 for variant N20D seems questionable when looking at the error bar.

-        L. 200: What is GGT?

-        Figure 1: Why not put the names of the variants immediately above the gel (vertically) instead of using the numbers that are harder to reference at a glance? Please put wt instead of BsNOX (which is BsNox everywhere else…) as a legend.

-        The reference to the former work does not belong in section 3.5 (ll. 218-221), but in the discussion.

-        Ll. 229-231: At which time point was this increase in 43 % observed?

Round 2

Reviewer 2 Report

The manuscript was strongly improved through the revision. A couple of minor points remain and should be addressed before publication:

-        The legends in figures 4 and 7 are strangely distorted, please correct.

-        Figure 8: Please elaborate on the figure legend. Especially for 8c and d it is unclear what is shown even with the help of the text.

-        The sentence “So we think that the structure’s quality is sufficient for our further work” (l. 181) should be written for a more scientific sound.

-        L. 293: very

-        Please check the newly written text in the discussion and the result part to remove repetitions.

-        The newly added author, WW, is annotated as having contributed equally to the two original first authors, but not mentioned in the author contributions.

Author Response

The manuscript was strongly improved through the revision. A couple of minor points remain and should be addressed before publication:

-        The legends in figures 4 and 7 are strangely distorted, please correct.

Response: Done and thanks for your good suggestion. (Please see the new legends in figures 4 and 7)

-        Figure 8: Please elaborate on the figure legend. Especially for 8c and d it is unclear what is shown even with the help of the text.

Response: We have elaborated the legends in figure 8. Thanks for your valuable suggestion.

-        The sentence “So we think that the structure’s quality is sufficient for our further work” (l. 181) should be written for a more scientific sound.

-        L. 293: very

Response: Revised and sorry for our misspellings.

-        Please check the newly written text in the discussion and the result part to remove repetitions.

Response: Revised and thanks for your good suggestion.

-        The newly added author, WW, is annotated as having contributed equally to the two original first authors, but not mentioned in the author contributions.

Response: Revised and thanks for your reminder.

Your efforts on reviewing this manuscript are greatly appreciated and once again, thank you so much for your time.